# KRAS and NRAS Translation Is Increased upon MEK Inhibitors-Induced Processing Bodies Dissolution

**DOI:** 10.3390/cancers15123078

**Published:** 2023-06-06

**Authors:** Olivia Vidal-Cruchez, Victoria J. Nicolini, Tifenn Rete, Karine Jacquet, Roger Rezzonico, Caroline Lacoux, Marie-Angela Domdom, Barnabé Roméo, Jérémie Roux, Arnaud Hubstenberger, Bernard Mari, Baharia Mograbi, Paul Hofman, Patrick Brest

**Affiliations:** 1Université Côte d’Azur, Institute of Research on Cancer and Aging of Nice (IRCAN), CNRS, INSERM, Centre Antoine Lacassagne, 28, Avenue de Valombrose, 06107 Nice, Francehofman.p@chu-nice.fr (P.H.); 2FHU-OncoAge, IHU-RESPIRera, 06001 Nice, France; 3Université Côte d’Azur, CNRS, INSERM, CNRS UMR7275, IPMC, 06560 Valbonne, France; 4Université Côte d’Azur, CNRS UMR7275, IPMC, 06560 Valbonne, France; 5Université Côte d’Azur, Institut Biologie Valrose (IBV), CNRS, Inserm, 06108 Nice, France; 6Université Côte d’Azur, CHU-Nice, Pasteur Hospital, Laboratory of Clinical and Experimental Pathology, Hospital-Integrated Biobank (BB-0033-00025), 06001 Nice, France

**Keywords:** liquid–liquid phase separation, MEK inhibitors, RAS oncogenic signaling, RNA metabolism, translation regulation

## Abstract

**Simple Summary:**

Cancer therapies directly targeting the mitogen-activated protein kinase (MAPK) pathway lead to cancer drug resistance. Resistance has been linked to compensatory RAS overexpression, but the mechanisms underlying this overexpression remain unclear. Here, we find that MEK inhibitors (MEKi) increases translation of the KRAS and NRAS oncogenes through a mechanism involving liquid–liquid phase separation (LLPS), and more particularly processing body (P-body) dissolution. Overall, we describe a new feedback loop mechanism involving P-bodies and phase separation that regulates RAS translation. Identification of key components that regulate LLPS will be important for future targeted therapeutic strategies.

**Abstract:**

Overactivation of the mitogen-activated protein kinase (MAPK) pathway is a critical driver of many human cancers. However, therapies directly targeting this pathway lead to cancer drug resistance. Resistance has been linked to compensatory RAS overexpression, but the mechanisms underlying this response remain unclear. Here, we find that MEK inhibitors (MEKi) are associated with an increased translation of the KRAS and NRAS oncogenes through a mechanism involving dissolution of processing body (P-body) biocondensates. This effect is seen across different cell types and is extremely dynamic since removal of MEKi and ERK reactivation result in reappearance of P-bodies and reduced RAS-dependent signaling. Moreover, we find that P-body scaffold protein levels negatively impact RAS expression. Overall, we describe a new feedback loop mechanism involving biocondensates such as P-bodies in the translational regulation of RAS proteins and MAPK signaling.

## 1. Introduction

Mitogen-activated protein kinases (MAPKs) are ubiquitous signal transduction pathways that control cell fate by phosphorylating hundreds of substrates. The RAS (KRAS, NRAS, HRAS)-RAF (ARAF, BRAF, RAF1)-MEK (MAP2K1/2)-ERK (MAPK1/3) pathway is altered in ~40% of all human cancers, mainly through oncogenic mutations in RAS (32%) and the downstream effector BRAF (~10%). KRAS is frequently mutated in pancreatic cancer (88%), lung cancer (30%), and colorectal adenocarcinomas (50%). NRAS and BRAF are altered in melanoma (17% and 55%, respectively), thyroid carcinoma (19% and 55%, respectively), and lung cancer (1% and 5%, respectively) [1,2,3,4]. Given the causative role of RAS-RAF-MEK-ERK pathway hyperactivation in tumorigenesis, several MEK, BRAF, and KRAS inhibitors (MEKi, BRAFi, and KRASi) have been developed in recent decades as potential therapeutic agents [5,6,7]. However, these treatments are invariably associated with drug tolerance, acquired resistance, and tumor recurrence [5,7,8]. Although some resistance can be directly attributable to the acquisition of somatic mutations, early drug resistance can also arise from drug-tolerant cells through transcriptional and post-transcriptional rapid reprogramming. Increased activation of the MAPK pathway and/or upregulation of KRAS and NRAS proteins have been suggested as possible non-genetic contributions that help cells evade KRASi, BRAFi, or MEKi treatment [5,9,10,11]. However, the molecular mechanism underlying overactivation of these proteins remains unknown.

Messenger RNAs (mRNA) are actively translated, repressed, stored, or degraded in response to environmental cues. These post-transcriptional regulatory mechanisms are critical for controlling cell fates that are altered under pathological conditions, such as cancer [12,13]. In this context, recent transcriptome-wide studies have shown that most RNAs have a restricted, dynamic, and regulated subcellular localization in biocondensates. The key players in this process include two cytoplasmic RNA clusters and their associated regulatory RNA-binding proteins, known as constitutive processing bodies (P-bodies) and environmentally induced stress granules [14,15]. In contrast to organelles with a lipid bilayer membrane, these membraneless structures are formed by a process known as liquid–liquid phase separation (LLPS), which confers a broad range of plasticity to these super-assemblies [16]. Sequestration of mRNAs into these biocondensates is associated with translational repression. These structures uncouple mRNA expression from protein production and enable spatiotemporal control of gene expression [14]. Moreover, P-bodies and stress granules act as reservoirs for silent mRNAs that can re-enter the translational pool and thus rapidly adapt protein levels to the environment and confer plasticity to the genetic program [14,16,17,18]. Although this post-transcriptional control is key to cellular adaptation to a versatile and stressful environment [12], few studies have investigated the role of P-bodies in the context of cancer progression, in particular in the context of the emergence of drug-tolerant cells [19].

In this study, we show that P-body dissolution participates in MEKi drug tolerance onset through a P-body-dependent negative feedback loop that fine-tunes the expression of KRAS and NRAS by controlling their mRNA translation.

## 2. Materials and Methods

### 2.1. Cell Culture

A549, Mel501, BT549 cells were cultured according to the recommendations of the ATCC. A549 (human lung adenocarcinoma epithelial cell line, LUAD, ATCC Cat# CCL-185, RRID:CVCL_0023) and Mel501 (melanoma epithelial cell line, SKCM, RRID:CVCL_4633, gift of Dr. Ballotti Laboratory) were grown in Dulbecco’s modified eagle medium (DMEM) supplemented with 10% fetal bovine serum (FBS). BT549 (Breast epithelial cell line, BRCA, ATCC ATCC Cat# HTB-122, RRID:CVCL_1092) was grown in DMEM supplemented with 10% FBS and non-essential amino-acid (Thermo Fisher Scientific, Illkirch-Graffenstaden, France). All the cells were maintained at 37 °C in a 5% CO_2_ in a humidified incubator. All cells were maintained for no more than one month and were identified using STR profiling (Eurofins Genomics, Ebersberg, Germany). For proliferation assays, 10,000 cells were seeded in six-well plates in triplicate and cell numbers were evaluated every day for 5 days using a Coulter counter (Beckman Coulter Life Sciences, Villepinte, France).

### 2.2. DDX6 GFP Stable Cell Lines

The pPRIPu GFP-DDX6 plasmid used in this study was constructed as follows: the pPRIPu CrUCCI vector [20] (kind gift of Dr F. Delaunay) was amplified with primer adaptors containing AgeI and BamHI restriction sites. pEGFP-C1_p54cp plasmid (kind gift of Drs D. Weil and M. Kress). The fragment was inserted in pPRIPU after digestion with AgeI and BamHI. The integrity of the entire sequence was confirmed by sequencing (Eurofins Genomics, Snapgene). Briefly, replication-defective, self-inactivating retroviral constructs were used to establish stable A549 cell lines. Selection was performed with puromycin (10 µg/mL). Cells were then sorted as a polyclonal population with homogeneous GFP expression and used in subsequent experiments (FACS Aria, IRCAN facility).

### 2.3. siRNA Transient Transfection

Cells were plated at 200,000 cells/well in 6-well plates. After 24 h, cells were transfected with negative control siRNA (silencer pre-designed small interfering RNA) or with pre-validated DDX6, LSM14A, EIF4ENIF1 (4E-T), KRAS, and/or NRAS siRNA using JetPrime (PolyPlus) according to the manufacturer’s instructions. Cells were lysed for RNA or protein analysis 48 h after transfection as described below. 

### 2.4. Cancer Cell Line Encyclopedia (CCLE) Expression Analyses

All CCLE data were obtained through DepMap Portal (https://depmap.org/portal/ (accessed on 1 October 2022)). The data were downloaded for LUAD, SKCM, and colorectal cancer (COAD) cell lines both for mRNA (n = 76, 78, 72, respectively) and Proteomics (n = 33, 28, 28, respectively). Pearson and Spearman coefficients and *p*-value were calculated with Prism8.0.2 program from GraphPad software.

### 2.5. Immunoblotting

Proteins were extracted from cells using Laemmli lysis buffer (12.5 mM Na_2_HPO_4_, 15% glycerol, 3% SDS). The protein concentration was measured with the DC Protein Assay (Bio-Rad, Gémenos, France) and 30 µg to 50 µg of total proteins were loaded onto 7.5% or 12% SDS-polyacrylamide gels for electrophoresis and transferred onto polyvinylidene difluoride membranes (Merck Millipore, Guyancourt, France). After 1 h of blocking with 3% bovine serum albumin or non-fat milk prepared in phosphate-buffered saline (PBS)-0.1% Tween-20 buffer, the blots were incubated overnight at 4 °C with primary antibodies (Appendix A). After 1 h of incubation with a horseradish peroxidase-conjugated secondary antibody (1:3000, Promega, Charbonnières-les-Bains, France), protein bands were visualized using an enhanced chemiluminescence detection kit (Merck Millipore) and the Syngene Pxi4 imaging system (Ozyme, Saint-Cyr-l’École, France). Immunoblot quantification was performed using Fiji freeware on unsaturated captured images.

### 2.6. Isolation of RNA and Quantitative Real-Time RT-PCR Analysis

Total RNA extraction was performed using TRI-reagent (Sigma-Aldrich, Merck Millipore, Guyancourt, France) to isolate the aqueous fraction, followed by column-based extraction (ZymoResearch, distributed by Ozyme, Saint-Cyr-l’École, France). The RNA concentration was measured via NanoDrop 2000 (Thermo Fisher Scientific, Illkirch-Graffenstaden, France). The cDNA strand was synthesized from 500 ng of total RNA (Thermo Fisher Scientific). Quantification of *KRAS*, *NRAS*, and *RPLP0* genes was measured via Power-Sybr-Green assays (Thermo Fisher Scientific) with the StepOne™ Real-time PCR System in agreement with The Minimum Information for Publication of Quantitative Real-Time PCR Experiments (MIQE) guidelines [21]. The qPCR primers are referenced in the Appendix A.

### 2.7. Immunofluorescence and Confocal Microscopy

Cells were grown to confluence and fixed in 4% paraformaldehyde for 15 min. After fixation, cells were permeabilized with a solution containing 0.3% Triton X-100 for 5 min and incubated with blocking buffer (0.03% Triton X-100, 0.2% gelatin and 1% BSA) for 20 min. Then, cells were incubated with primary antibodies (Appendix A) overnight at 4 °C in humidified chambers in blocking buffer. Cells were washed and incubated with Alexa Fluor-conjugated secondary antibodies (1:500; Thermo Fisher Scientific) for 1 h at room temperature, washed, and mounted in ProLong Diamond Reagent with DAPI (Thermo Fisher Scientific). Images were captured on a Zeiss LSM880 confocal microscope (Plan-Apochromat 40×/1.4 Oil DIC M27) and analyzed with Fiji freeware.

### 2.8. Polysome Gradient

All steps of the subcellular fractionation were performed at 4 °C. Cells (60–80 × 10^6^ cells) with or without treatment were trypsinized and washed twice with 15 mL of ice-cold PBS by centrifugation. Cell pellets were resuspended in 1 mL of hypotonic buffer composed of 40 mM Tris-HCl, pH 7.5, 10 mM KCl, 3 mM MgCl_2_ and 0.2% Nonidet P-40 supplemented with 1 mM DTT and 0.5 mM phenylmethylsulfonyl fluoride (PMSF). After incubation for 15 min on ice, cells were lysed in a Dounce homogenizer with B pestle. The efficiency of cell lysis in keeping the nuclei intact was verified by staining with trypan blue under an optical microscope. The cell lysates were spun at 900× *g* for 5 min to obtain the cytoplasm (supernatant) and nuclei (pellet) fractions.

To perform the sucrose gradient fractionation, about 10 OD 260 nm of cytosolic extracts were loaded onto a linear sucrose gradient (10–50%) made of sucrose gradient buffer (25 mM Tris-HCl, pH 7.5, 150 mM NaCl, 12 mM MgCl_2_, 1 mM DTT) in Ultra-Clear ultracentrifugation tubes (Beckman Coulter Life Sciences). The cytosolic fraction samples were fractionated via ultracentrifugation for 2 h 45 min at 39,000 rpm and at 4 °C in a Beckman Optima ultracentrifuge with a SW41 Ti swinging bucket rotor at the following settings: acceleration 9 and deceleration 4. Following ultracentrifugation, the sucrose gradients were fractionated using a Foxy JR fraction collector and monitored using a UV light (254 nm wavelength) absorbance detector (Teledyne ISCO, UA-6) to obtain 12 to 14 fractions. After addition of 0.5 mM CaCl_2_ and 0.2% SDS, the collected fractions were treated with proteinase K (50 mg/mL, Sigma-Aldrich, Merck Millipore) for 30 min at 40 °C. Then, 10 pg of an RNA spike-in was added to each fraction to serve as an internal calibrator of RNA extraction and detection. For this, we used an in vitro transcribed RNA that encodes the spike protein of SARS-CoV-2.

Total RNA was extracted via vigorous shaking with an equal volume of Tris pH 8 saturated phenol, chloroform, and isoamyl alcohol mixture (25:24:1, *v/v/v*) (Sigma-Aldrich, Merck Millipore), and phase separation was performed via centrifugation at 12,000× *g* for 15 min at 4 °C. The upper aqueous phase was washed once with an equal volume of chloroform:isoamyl alcohol (24:1, *v/v*) (Sigma-Aldrich, Merck Millipore) via vigorous shaking and centrifugation at 12,000× *g* for 15 min at 4 °C. The aqueous phase was transferred to a fresh tube and the RNA was precipitated overnight at −20 °C by mixing with an equal volume of isopropanol, 2 mL of glycoblue TM co-precipitant (Thermo Fisher Scientific), and 1/10 volume of 3 M Sodium acetate pH 5.2. RNA was pelleted by centrifugation at 12,000× *g* for 15 min at 4 °C. 

### 2.9. Statistical Analysis

Quantitative data were described and presented graphically as medians and interquartiles or means and standard deviations. The distribution normality was tested with Shapiro’s test and homoscedasticity with a Bartlett’s test. For two categories, statistical comparisons were performed using the Wilcoxon matched-pairs signed rank test or Mann–Whitney’s test. All statistical analyses were performed by the biostatistician using R.3.2.2 software and the Prism8.0.2 program from GraphPad software. Tests of significance were two-tailed and considered significant with an alpha level of *p* < 0.05. (graphically: * for *p* < 0.05, ** for *p* < 0.01, *** for *p* < 0.005).

## 3. Results

### 3.1. MEKi Response Promotes KRAS and NRAS Translation

In order to confirm that NRAS and KRAS proteins accumulate upon MEK pathway inhibition, we treated three different cell lines (A549, Mel501, BT549) with MEKi, PD184352 (CI-1040), and trametinib. These cell lines, which included a range of cancer types (lung, melanoma, or breast) and oncogenic drivers (KRASG12S, BRAFV600E, and PTEN, respectively) [22], showed accumulation of NRAS and KRAS under MEKi treatments (Figure 1A–C). 

We next investigated the possible causes for this increase in KRAS and NRAS protein expression. mRNA levels of KRAS and NRAS were unaffected by MEKi treatment (Figure 2A,B). These data argue that increased transcription does not account for increased levels of KRAS and NRAS proteins. However, ribosome profiling indicated that upon MEKi treatment, a part of the mRNAs of KRAS and NRAS shifted from monosomes to heavy polysome fractions associated with increased translation (Figure 2C,D). Altogether, these results show that KRAS and NRAS levels are regulated through a post-transcriptional mechanism whose translation inhibition is withdrawn upon MEKi treatment.

### 3.2. MEK Inhibition Promotes P-body Dissolution

KRAS and NRAS mRNAs have been shown to be accumulated in P-bodies [14]; thus, we hypothesized that overexpression of KRAS and NRAS proteins should inversely correlate with P-bodies. We therefore analyzed the number and size of cellular P-bodies in different cancer cell lines. Strikingly, we observed a significant decrease in the size and number of P-bodies in response to PD184352 and trametinib in all cell types examined (Figure 3A–E and Appendix A). This MEKi induced a significant P-body dissolution after 8 h. The KRAS and NRAS level changes were progressive, with a full increase after 24 h. (Figure 3F–H and Appendix A). MEKi had no effect on cell cycle progression at the 8 h time point, arguing that this mechanism precedes cell cycle arrest, rather than being a consequence of it (Appendix A). Furthermore, the overall decrease in the number of P-bodies was due to protein relocation rather than loss of expression of DDX6 or LSM14A [14,23,24], two scaffold proteins required for P-body formation (Figure 3H). In addition, we analyzed stress granules, a second type of cytoplasmic membraneless organelles, using G3BP1 staining to observe if the mRNA were relocated in those granules. As we could not witness any stress granules induction, we concluded that this relocation was cytoplasmic, rather than in favor in other membraneless granules (Appendix A). Altogether, these converging results show that the dissolution of P-bodies under MEKi treatment correlates with RAS increased expression.

### 3.3. Relationship between KRAS and NRAS Expression and P-body Components

We analyzed the contribution of essential P-body components on RAS expression independently of MEKi treatment. To test for a causal link between the translational repression activities of P-body components and KRAS and NRAS expression, we modulated the level of the RNA helicase DDX6, an essential P-body component [23]. Overexpression of GFP-tagged DDX6 increased the size and number of P-bodies. This significant increase in P-bodies correlated with decreased KRAS and NRAS expression (Figure 4A–C and Appendix A). Notably, in these conditions, mRNA levels of KRAS and NRAS remained comparable, suggesting a mechanism involving mRNA storage and translation inhibition (Figure 4D,E). The silencing of DDX6, LSM14, and 4E-T, which resulted in dissolution of P-bodies [23], was correlated with a slight increased RAS expression (Figure 5 and Appendix A) but also associated with important lethality that might play an indirect role on RAS expression (Appendix A). 

To further investigate the relationship between the components of the P-body and the expression of RAS, we analyzed their correlation in different cell lines thanks to the CCLE database, focusing on LUAD and SKCM cancers. While no correlation was found between DDX6 and NRAS mRNA in LUAD cell lines, we observed a significant inverse correlation between DDX6 and NRAS proteins (Figure 6A,B), confirming our previous results. In addition, we also detected a negative correlation between LSM14A and NRAS protein in LUAD and no correlation at the mRNA level (Figure 6C,D). The inverse correlation between DDX6 and NRAS was also observed in SKCM (Figure 6E,F). Significant inverse correlations between DDX6 and KRAS, 4E-T and KRAS, and DDX6 and NRAS were also observed in colorectal adenocarcinoma (COAD) cell lines (Appendix A). Altogether, these results showed that some P-body components can regulate the translation of RAS proteins in different adenocarcinoma cell types. 

### 3.4. MAPK Signaling Promotes P-body Formation

Since the size and number of P-bodies are intricately associated with KRAS and NRAS expression, we investigated the consequences of MEKi treatment removal both on P-body regulation and MAPK signaling. We therefore treated cells with MEKi for 24 h to reduce P-body numbers and induce KRAS and NRAS overexpression. MEKi was then washed out and cells were harvested at different time points (Figure 7A). Directly after MEKi washout, a strong activation of ERK was observed that persisted for 4 h, along with the stable expression of KRAS and NRAS and phosphorylation of BRAF (Figure 7B). From 8 to 24 h, a gradual decrease in KRAS and NRAS expression along with a decrease in BRAF phosphorylation was inversely correlated with an increasing number of P-bodies over time (Figure 7B–D and Appendix A), strongly suggesting a translational arrest. Removal of the drug resulted in re-establishment of a growth rate comparable to that of untreated cells, demonstrating the adaptability of cancer cells to drug treatment cycles (Figure 7E). Our results illustrate the high plasticity orchestrated by and the role of P-bodies to fine-tune the translational rate of RAS proteins through negative feedback loop. 

### 3.5. Expression of KRAS or NRAS Is Sufficient to Induce Phosphorylation of BRAF

Furthermore, to analyze whether both KRAS and NRAS proteins activate downstream BRAF phosphorylation under these conditions, we used siKRAS and siNRAS either alone or in combination (Figure 8A,B). We demonstrated that phosphorylated BRAF accumulates in the presence of MEKi. Treatment with either siKRAS or siNRAS led to partial inhibition of BRAF phosphorylation, while complete inhibition of BRAF phosphorylation was only achieved when both siRNAs were combined (Figure 8A–C). Altogether, these results highlight that increasing protein expression of KRAS or NRAS was sufficient to induce phosphorylation of BRAF after MEKi and P-body dissolution.

## 4. Discussion

For many decades, studies have pointed to genetic mutations as a central mechanism for acquiring resistance to targeted therapies [25]. However, there is a growing body of evidence that challenges this consensus and suggests that non-genetic heterogeneity and cell plasticity are actively involved in drug tolerance [26,27,28]. Recently, new profiling techniques such as FATE-seq have helped to unravel the non-genetic mechanisms of resistance [29]. Nevertheless, genetic and non-genetic mechanisms of resistance or drug tolerance are often interrelated and not mutually exclusive [28,30]. Non-genetic resistance is due to the intrinsic plasticity of tumor cells, i.e., the ability to undergo transcriptional and epigenetic reprogramming in response to environmental challenges or to therapy. In this context, current therapeutic options for BRAFV600E/K patients include therapies targeting the MAPK pathway, which show remarkable efficacy in the first months of treatment [8]. However, most patients treated with a combination of KRASi, BRAFi, and MEKi inevitably relapse within a few months [8,31]. This relapse is associated with the presence of so-called persister, cancer stem, drug tolerant, or resistant cells as reported in several studies, that harbor either a genetic or non-genetic program that supports survival despite drug treatment [27,32].

Here, we focused on the early events of drug tolerance. We demonstrated that cells were able to establish overexpression of KRAS and NRAS that was independent of de novo transcription. Continuous versus intermittent BRAF and MEK inhibition in melanoma patients with BRAFV600E/K mutations was tested in a randomized, open-label phase 2 clinical trial (NCT02196181) [31]. In this clinical trial, continuous administration resulted in a statistically significant improvement in progression-free survival after randomization compared to intermittent administration. These results suggest that the attempt to counteract drug-associated genetic acquisition of resistance is detrimental if drug-tolerant cancer cells with greater plasticity grow faster and thus increase the number of persister cells after each cycle of BRAFi/MEKi treatment. Furthermore, resistance to KRASG12C inhibitors involves MAPK pathway overactivation and patients harboring either KRAS or NRAS de novo mutations [6,11,33]. Recently, BI-3406, a SOS1–KRAS interaction inhibitor, was shown to attenuate SOS1 dependent feedback reactivation induced by MEK inhibitors and thereby enhance sensitivity of KRAS-dependent cancers to MEK inhibition [34]. Our results here demonstrate another upstream feedback loop, revealing a new approach to potentially targeting this pathway by preventing both KRAS and NRAS translation or activation.

We have also shown that both PD184352 and trametinib can decrease the size and number of P-bodies. The positive effect of the MAPK pathway on P-bodies is consistent with a previous siRNA screen of 1354 human genes which revealed that MAPK3 (ERK1) was associated with P-body dissolution in the absence of a DDX6 decrease [35]. Our work highlights the fact that the activity, rather than the level, of ERK is critical for P-body dissolution. However, since ERKs can phosphorylate more than 200 intracellular targets [36], we investigated whether one of these targets was part of a protein essential for P-body composition [14]. While LSM14A appears to be a putative indirect target of ERK in rats, the ERK phosphorylation motif is not conserved in human. For EIF4ENF1 (4E-T), a direct site is present in mouse, while an indirect site is present in humans [36]. In addition, our experimental data combined with bioinformatics analysis of many cell lines showed an inverse correlation between P-body protein components and RAS translation. In the siRNA screen of 1354 human genes cited earlier, some invalidations were found to be associated with P-body dissolution and/or DDX6 downregulation [35], so it would be interesting to identify the limiting component in the different cell lines or tissues. Several features of KRAS and NRAS mRNAs (i.e., the length of 5.3 kb and 4.3 kb, respectively, and the low GC content of their coding region or 3’UTR mRNA, 38% and 44%, respectively) may also contribute to their targeting of P-bodies [37]. Previous studies have shown that KRAS is poorly translated due to its enrichment in genomically underrepresented or rare codons [38,39]. In our study, we observed that MEKi-induced dissolution of P-bodies was associated with strong translation of KRAS and NRAS mRNAs and overexpression of the corresponding proteins. These results suggest that rare codons, often associated with the presence of an AU-rich third codon, favor the recruitment of mRNA repressors due to slow translational processing, which, however, can be overcome under certain conditions, such as MEKi treatment. 

Finally, the accessibility of the mRNA was modified due to the dissolution of P-bodies. Previously, MEKi, in association or not with BRAFi, have been shown to induce an important change in translation by favoring the expression of specific proteins through the activation of the eIF4 complex [40,41]. Here, we showed that the monosomal fraction was reduced in favor of both the unbound and the polysomal fraction. Furthermore, the change in translation was described to be even more potent in persister cells, showing that redistribution mRNA and their increased accessibility, coupled with selective translation, may contribute to the adaptative plasticity of cancer cells to treatments [40,41].

## 5. Conclusions

Overall, these results reveal a novel negative feedback loop involving P-bodies in the translational control of KRAS and NRAS mRNAs. The type of regulation of the MAPK pathway that we have described here should pave the way for therapies that avert early resistance and target drug-tolerant cells before relapse.

## Figures and Tables

**Figure 1 cancers-15-03078-f001:**
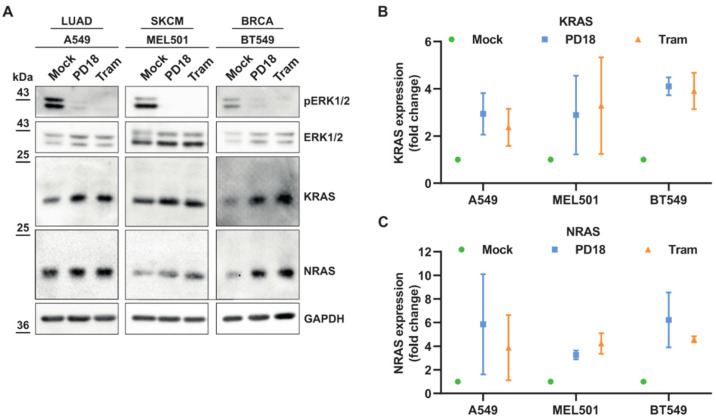
MEKi treatments induce potent KRAS and NRAS overexpression. (**A**). Cancer cells were treated for 24 h with PD184352 (PD18) and trametinib (Tram) at 10 µM and 10 nM, respectively. LUAD: lung adenocarcinoma; SKCM: skin cutaneous melanoma; BRCA: breast cancer. Western blot analysis of the indicated proteins. pERK1/2 represent phosphorylated forms of ERK1/2. Results are representative of independent experiments. Original Western blots can be seen in Appendix A. (**B**,**C**) Quantification of KRAS and NRAS expression, respectively by combining A549 (n = 3), MEL501 (n = 2), and BT549 (n = 2) biological replicates upon indicated treatments.

**Figure 2 cancers-15-03078-f002:**
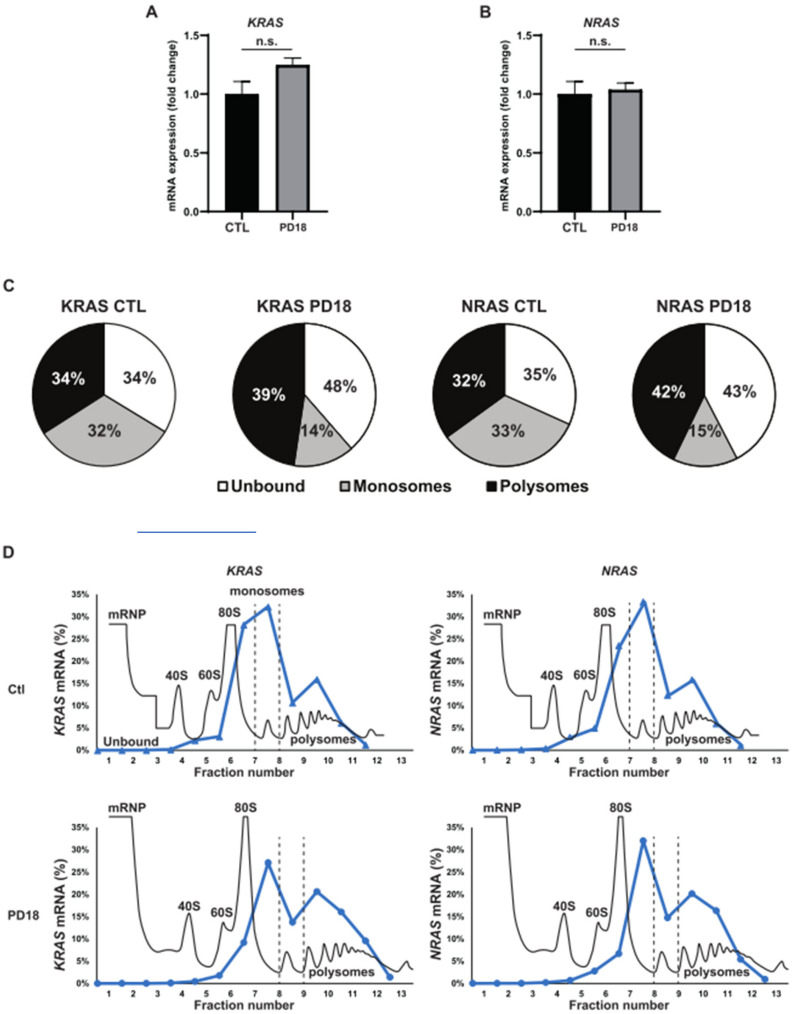
Translation-dependent induction of KRAS and NRAS overexpression by MEKi treatments. (**A**). A549 cells were subjected to a 24 h treatment with PD184352 (PD18). The RNA expression of KRAS mRNA was analyzed through RT-qPCR experiments, with RPLP0 serving as the housekeeping gene. (**B**). Similarly, the RNA expression of NRAS under the same conditions was examined, with RPLP0 used as the housekeeping gene. (**C**). A549 cells were treated with PD184352 (PD18) at a concentration of 10 µM for 24 h. The quantification of KRAS and NRAS mRNA bound to different fractions (unbound, monosomes, polysomes) obtained from polysome fractionation through sucrose-gradient ultracentrifugation was conducted. Total mRNA was normalized to 100%. All fractions preceding the monosome fraction were designated as “unbound”, while all fractions subsequent to the monosome fraction were designated as “polysomes”. (**D**). The figure presents the polysome profiles (continuous black line) of one representative profile from two independent experiments. The dashed lines indicate the separation boundaries between the “unbound”, “monosomes,” and “polysomes” fractions. The RNA levels (continuous blue line) of KRAS and NRAS in each fraction were quantified using RT-qPCR and normalized by the SARS-CoV-2 spike-in gene, as described in the methods section.

**Figure 3 cancers-15-03078-f003:**
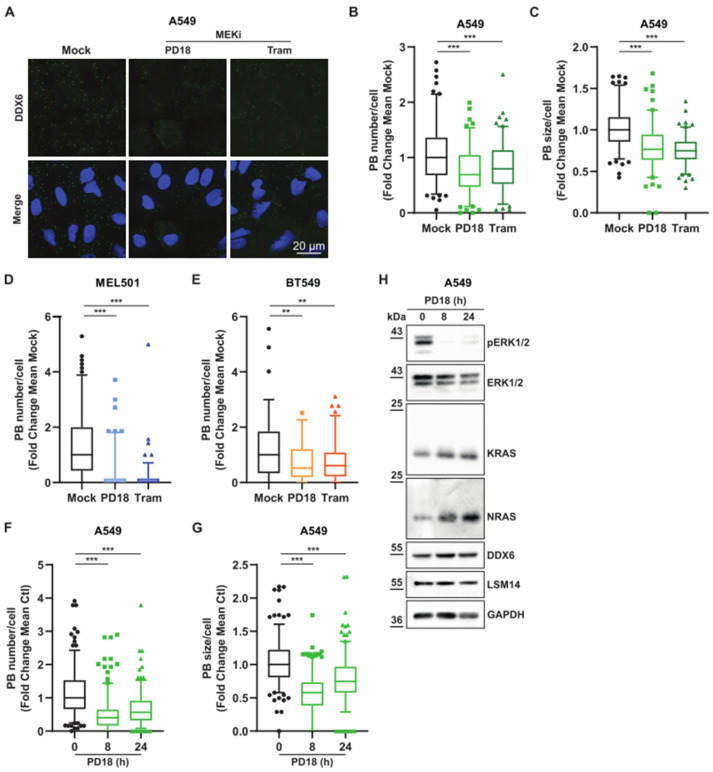
MEKi treatment is associated with a decrease in P-body size and number. (**A**). A549 cells were treated 24 h with PD184352 (PD18) and trametinib (Tram) at 10 µM and 10 nM, respectively. Confocal analysis of P-bodies using anti-DDX6 antibodies (green) with DAPI nuclear staining (blue). Results are representative of 3 independent experiments. (**B**,**C**) Quantification of indicated P-body parameters of previous experiments by FIJI. Results of 3 independent and merged experiments with a minimum of 120 cells in total per condition. A Mann–Whitney test was performed for statistical analysis. *p*-values are indicated as, ** < 0.01, *** < 0.005. (**D**,**E**). MEL501, BT549 cells were treated for 24 h with PD184352 (PD18) and trametinib (Tram) at 10 µM and 10 nM, respectively. Quantification of indicated P-body parameters by FIJI. Results are representative of 3 independent merged experiments with a minimal quantification of 70 cells in total per condition. A Mann–Whitney test was performed for statistical analysis. *p*-values are indicated as, ** < 0.01, *** < 0.005. (**F**–**H**) A549 cells were treated with PD184352 (PD18) at 10 µM and harvested at the indicated time. (**F**,**G**). Quantification of indicated P-body parameters by FIJI. Results are representative of 3 independent merged experiments with a minimal quantification of 220 cells in total per condition. A Mann–Whitney test was performed for statistical analysis. *p*-values are indicated as, ** < 0.01, *** < 0.005. (**H**) Western blot analysis at the indicated time. pERK1/2 represents a phosphorylated form of ERK1/2. The results are representative of 2 independent experiments. Original Western blots can be seen in Appendix A.

**Figure 4 cancers-15-03078-f004:**
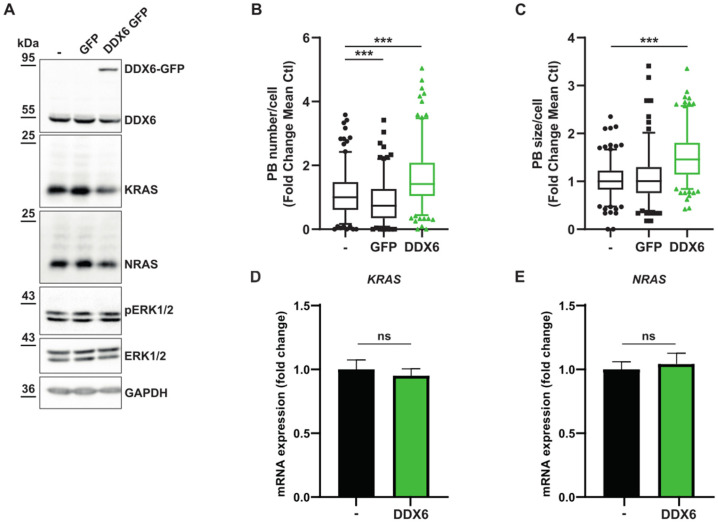
DDX6 overexpression decrease KRAS and NRAS translation. (**A**–**E**). Populations of A549 cells either untreated (−), or overexpressing a green fluorescent protein (GFP), or a DDX6-GFP fused protein (DDX6-GFP). (**A**). Western blot analysis of indicated proteins. The results are representative of 2 independent experiments. Original Western blots can be seen in Appendix A. (**B**,**C**) Quantification via FIJI of P-body number or size, respectively. Results are representative of 3 independent merged experiments with a minimal quantification of 160 cells for each condition. A Mann–Whitney test was performed for statistical analysis. *p*-values are indicated as *** < 0.005. (**D**,**E**) *KRAS* and *NRAS* mRNA expression analysis of the indicated genes was analyzed by RT-qPCR using *RPLP0* for normalization and untreated (−) as reference. Results represent the combination of 3 independent experiments. A Mann–Whitney test was performed for statistical analysis (n.s.: non-significative).

**Figure 5 cancers-15-03078-f005:**
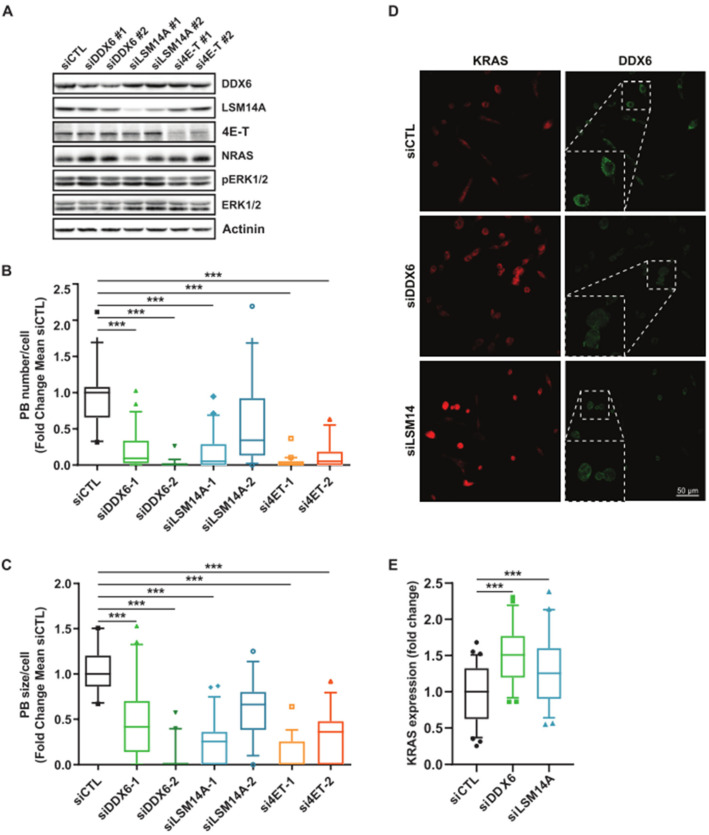
Pbodies components invalidation increase KRAS and NRAS expression. (**A**). A549 cells were transfected with a siControl (siCTL) or with siDDX6, LSM14, or 4E-T for 48 h. Western blot analysis of indicated proteins. Results are representative of 3 independent experiments. Original Western blots can be seen in Appendix A. (**B**,**C**). Quantification via FIJI of P-body number in the same conditions. Results are representative of 3 independent merged experiments with a minimal quantification of 110 cells in total per condition. A Mann–Whitney test was performed for statistical analysis. *p*-values are indicated as *** < 0.005. (**D**). A549 cells were transfected with a siControl (siCTL), siDDX6, LSM14. Cells were harvested at 48 h post transfection and stained with KRAS (Red) and DDX6 (Green) antibodies. (**E**). Total immunofluorescence of KRAS by cell of z-stacked images was quantified. A Mann–Whitney test was performed for statistical analysis. *p*-values are indicated as *** < 0.005.

**Figure 6 cancers-15-03078-f006:**
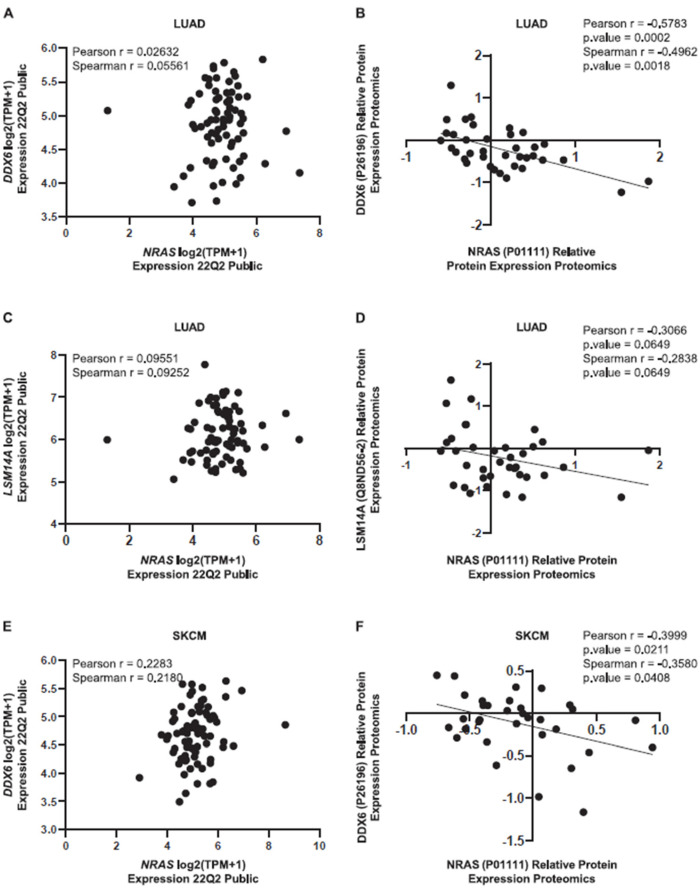
P-body protein levels inversely correlate with NRAS expression levels. (**A**). *NRAS* mRNA levels (n = 76) relative to *DDX6* mRNA levels in LUAD cell lines from CCLE database (Depmap portal). (**B**). NRAS protein levels (n = 33) relative to DDX6 protein levels in LUAD cell lines. (**C**). *NRAS* mRNA levels (n = 76) relative to LSM14A mRNA levels in LUAD cell lines. (**D**). NRAS protein levels (n = 33) relative to LSM14A protein levels in LUAD cell lines. (**E**). *NRAS* mRNA levels (n = 78) relative to *DDX6* mRNA levels in SKCM cell lines. (**F**). NRAS protein levels (n = 28) relative to DDX6 protein levels in SKCM cell lines.

**Figure 7 cancers-15-03078-f007:**
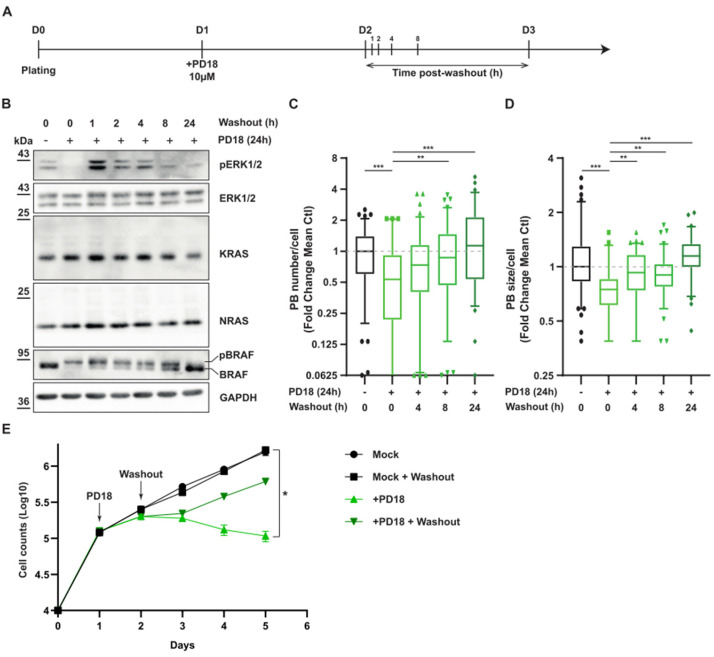
Dynamic regulation of P-body formation and MAPK signaling. A549 cell were treated with PD184352 (PD18) at 10 µM. After 24 h MEKi was removed, and cells were harvested at the indicated time. (**A**). Scheme of the experiments presented in panel (**B**–**D**). (**B**). Western blot analysis at the indicated time. pERK and pBRAF represent the phosphorylated forms of ERK and BRAF, respectively. The results are representative of 2 independent experiments. Original Western blots can be seen in Appendix A. (**C**,**D**). Quantification via FIJI of P-body number and size, respectively. Results are representative of 2 independent merged experiments with a minimal quantification of 60 cells in total per condition. A Mann–Whitney test was performed for statistical analysis. For all experiments, *p*-values are indicated as ** < 0.01; *** < 0.005. (**E**). Cell counts at indicated time. Results are representative of 6 independent biological replicates, * < 0.05.

**Figure 8 cancers-15-03078-f008:**
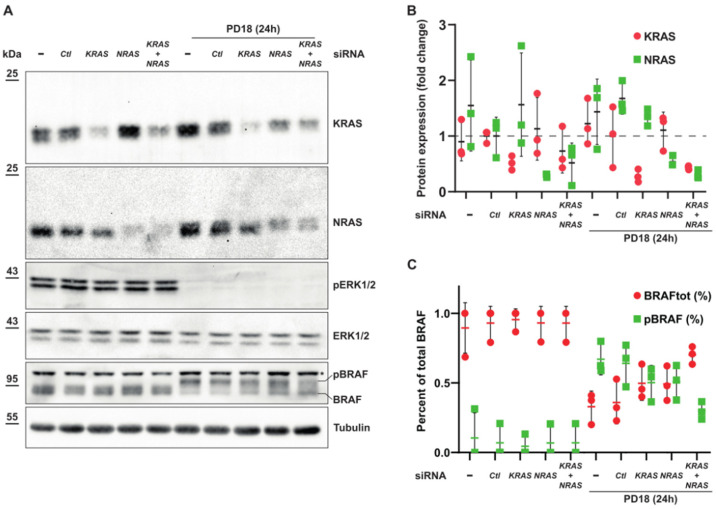
KRAS and NRAS can have compensatory effect to phosphorylate BRAF. (**A**). A549 cells were transfected with the indicated siRNAs for 24 h followed by 24 h of treatment with PD184352 at 10 µM (PD18). (*) unspecific band. Original Western blots can be seen in Appendix A. (**B**). Quantification of KRAS and NRAS proteins normalized using siCTL condition (n = 3 independent biological experiments, mean+/− s.d.). (**C**). Quantification of BRAF forms as a percentage of total BRAF protein (n = 3 independent biological experiments, mean+/− s.d.).

## Data Availability

All CCLE data were obtained through DepMap Portal (https://depmap.org/portal/ (accessed on 1 October 2022)).

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
