# Peer review of "KRAS and NRAS Translation Is Increased upon MEK Inhibitors-Induced Processing Bodies Dissolution"

_cancers, 2023, doi:10.3390/cancers15123078_

Round 1

Reviewer 1 Report (New Reviewer)

In this study, the authors present some interesting findings that the MEKi-induced overexpression of RAS proteins is associated with the dissolution of P-bodies. They demonstrate that the increase in KRAS and NRAS protein levels is not due to an increase in mRNA level, but rather an increase in translation, as evidenced by the presence of more KRAS and NRAS mRNAs in the polysomes (actively translated). Correlatively, they find that the size and number of P-bodies decrease in different cancer cell lines upon MEKi treatment, and overexpression of the P-body component DDX6 decreases KRAS and NRAS protein levels but not mRNA levels. The authors attempt to dissolve P-bodies by knocking down key components of P-bodies (DDX6, LSM14A, 4E-T), which should lead to an increase in KRAS and NRAS protein levels. However, the data is not very strong in support of this point. Overall, the findings of this study are interesting and support some of the known functions of P-bodies, such as mRNA storage. They also increase our knowledge of LLPS and cancer.

I believe this paper is publishable. However, there are several major points the authors should address and clarify before that, especially since there are some inconsistencies with the data presented in this work.

Firstly, the percentage numbers in figure 2C and 2D do not align (the numbers for free fractions are all close to 0 in figure 2D). Moreover, the authors should annotate which fractions were used as free, monosomes, and polysomes, respectively. A more detailed explanation of how the data was processed and normalized should be provided in either the legend or the methods section. Additionally, the control mRNA used for RT-qPCR in Figure 2A needs to be defined.

Secondly, the data shown in figure S2 is not consistent with the data shown in figure 3A: treatment of PD18 does not decrease the P-body number from the DDX6 signal. Moreover, the statement "This relocation was cytoplasmic rather than in favor of stress granules induction" is confusing. The authors need to explain the rationale more clearly.

Thirdly, why is the WB for KRAS missing from figure 5A?

Fourthly, the authors describe this P-body-dependent KRAS and NRAS expression regulation as a negative feedback loop. Could the authors draw a scheme to clarify that?

Some minor comments:

There are several places where references are missing (lines 197-201, 250, 277, and 282-285).

Figure 1A: Why are there two WB bands for ERK proteins? Are they both ERK proteins? Please annotate that in the legend.

Line 220: This conclusion needs to be modified to "whose translation inhibition is withdrawn upon MEKi treatment."

Figure 8A: In the BRAF WB panel, please annotate what "*" means.

Some of the sentences in the author's writing may be challenging to read due to a lack of clear introduction or lengthy sentence structure.

Author Response

In this study, the authors present some interesting findings that the MEKi-induced overexpression of RAS proteins is associated with the dissolution of P-bodies. They demonstrate that the increase in KRAS and NRAS protein levels is not due to an increase in mRNA level, but rather an increase in translation, as evidenced by the presence of more KRAS and NRAS mRNAs in the polysomes (actively translated). Correlatively, they find that the size and number of P-bodies decrease in different cancer cell lines upon MEKi treatment, and overexpression of the P-body component DDX6 decreases KRAS and NRAS protein levels but not mRNA levels. The authors attempt to dissolve P-bodies by knocking down key components of P-bodies (DDX6, LSM14A, 4E-T), which should lead to an increase in KRAS and NRAS protein levels. However, the data is not very strong in support of this point. Overall, the findings of this study are interesting and support some of the known functions of P-bodies, such as mRNA storage. They also increase our knowledge of LLPS and cancer.

I believe this paper is publishable. However, there are several major points the authors should address and clarify before that, especially since there are some inconsistencies with the data presented in this work.

We thank the reviewer for the positive comments, we have now taken into account all the suggestions in order to improve the quality of the manuscript.

Firstly, the percentage numbers in figure 2C and 2D do not align (the numbers for free fractions are all close to 0 in figure 2D). Moreover, the authors should annotate which fractions were used as free, monosomes, and polysomes, respectively. A more detailed explanation of how the data was processed and normalized should be provided in either the legend or the methods section. Additionally, the control mRNA used for RT-qPCR in Figure 2A needs to be defined.

All the modifications have been done accordingly to the reviewer comments.

The added dash lines in the figure 2D in order to explain our quantifications in Figure 2C. We added a sentence in the legend of the figure to explain that the mRNA was normalized using a spike in (SARS cov2) and a link to the method section.

The added in the figure legend 2A and B that we used RPLP0 gene as housekeeping gene for normalization.

Secondly, the data shown in figure S2 is not consistent with the data shown in figure 3A: treatment of PD18 does not decrease the P-body number from the DDX6 signal. Moreover, the statement "This relocation was cytoplasmic rather than in favor of stress granules induction" is confusing. The authors need to explain the rationale more clearly.

We have changed the figure of PD18 and improved the contrast. Furthermore, we added a part in the text to explain the rationale: “In addition, we analyze stress granules, a second type of cytoplasmic membraneless organelles, using G3BP1 staining to observe if the mRNA were relocated in those granules. As we could not witness any stress granules induction, we concluded that this relocation was cytoplasmic rather than in favor in other membraneless granules (Supplementary Fig. S2)”.

Thirdly, why is the WB for KRAS missing from figure 5A?

Unfortunately, we are experiencing troubles with the latest AC batch: It is functioning well in IF but not in WB. I regret to inform you that we cannot provide any further anti-KRAS WB at this time.  I apologize for the inconvenience

 Fourthly, the authors describe this P-body-dependent KRAS and NRAS expression regulation as a negative feedback loop. Could the authors draw a scheme to clarify that?

A graphical abstract was submitted separately of the main document. We now added it directly in the manuscript.

Some minor comments:

There are several places where references are missing (lines 197-201, 250, 277, and 282-285).

This has been corrected.

Figure 1A: Why are there two WB bands for ERK proteins? Are they both ERK proteins? Please annotate that in the legend.

This has been corrected.

Line 220: This conclusion needs to be modified to "whose translation inhibition is withdrawn upon MEKi treatment."

This has been corrected.

Figure 8A: In the BRAF WB panel, please annotate what "*" means.

The star was meaning unspecific band. This has been specified in the figure legend.

Comments on the Quality of English Language:

Some of the sentences in the author's writing may be challenging to read due to a lack of clear introduction or lengthy sentence structure.

We are deeply sorry to hear that since we have previously asked for 2 independent English reviewing by native speakers.

Reviewer 2 Report (New Reviewer)

The authors study the effect of the processing bodies dissolution or formation on KRAS and NRAS expression, starting from the point that overexpression of RAS genes correlates with tumour resistance. They found that mechanism bringing to P-body dissolution leads to increased translation of NRAS e KRAS; on the other hand when the number of P-bodies increases there is reduced RAS-signalling. They shows that the current therapies which use MEKi or other similar drugs targeting MAPK pathway show some resultes at the beggining but cause relapse within a few months. According to this, the challenge is to find a combination therapy that helps to overcome resistance acquisition. They discovered a negative feedback loop involving P-Bodies and suggest a new type of regulation of the MAPK pathway.

Author Response

We thank the reviewer for the positive comments.

Round 2

Reviewer 1 Report (New Reviewer)

Most of my concerns has been nicely addressed. 

My only point is regarding Figure 2C and 2D: I could not see the "added dash line" in the revise manuscript. I am still confused how one transforms the number from Figure 2D to Figure 2C. Are they using the same unit? Did the author simply add the numbers from the blue dots in Figure 2D to get the number in Figure 2C or there is a further normalization step? Please clarify. 

Author Response

We apologize for any confusion caused. The dashed lines in Figure 2D were specifically included to delineate the "monosomes" fractions from the others. However, to enhance clarity and improve understanding, we have now supplemented the figure legend with additional sentences.

This manuscript is a resubmission of an earlier submission. The following is a list of the peer review reports and author responses from that submission.

Round 1

Reviewer 1 Report

The manuscript by Vidal-Cruchez and colleagues makes an interesting and novel contribution to our understanding of the important issue of resistance to signal transduction inhibitors. The writing is clear and concise and the experiments designed well and reported in extensive figures that have useful statistical analyses.  Overall, only a few minor improvements are suggested:

1. Title (line 2): use of "improved" seems wrong because it implies a value judgement. Better to just say "Increased" or "Enhanced"?

2. Title and Simple Summary (lines 2 and 24): should probably define "MEKi" to help if those are viewed separately.

3. Would be useful to the reader to state the dilutions of the primary antibodies used for the immunoblotting (line 127) and immunofluorescence (line 146). Probably best to add that information to supplementary table 1.

4. "106" in line 153 probably stands for million and shows loss of the superscript.  Loss of subscript and superscript numbers is a problem at various places in the manuscript.

Author Response

Reviewer 1 :

The manuscript by Vidal-Cruchez and colleagues makes an interesting and novel contribution to our understanding of the important issue of resistance to signal transduction inhibitors. The writing is clear and concise and the experiments designed well and reported in extensive figures that have useful statistical analyses.  Overall, only a few minor improvements are suggested:

We thank the reviewer for the positive comments, we have now taken into account all the suggestions in order to improve the quality of the manuscript.

  1. Title (line 2): use of "improved" seems wrong because it implies a value judgement. Better to just say "Increased" or "Enhanced"?

This has been corrected.

  1. Title and Simple Summary (lines 2 and 24): should probably define "MEKi" to help if those are viewed separately.

This has been corrected.

  1. Would be useful to the reader to state the dilutions of the primary antibodies used for the immunoblotting (line 127) and immunofluorescence (line 146). Probably best to add that information to supplementary table 1.

This has been corrected.

  1. "106" in line 153 probably stands for million and shows loss of the superscript.  Loss of subscript and superscript numbers is a problem at various places in the manuscript.

This has been corrected.

Reviewer 2 Report

This study is to investigate whether the induction of RAS proteins by the MEK inhibitors (MEKi) is associated with a change in Processing Body (P-body) biocondensates in cancer cells. The suthors showed that both PD184352 and trametinib can decrease the size and number of P-Bodies, which indicating that P-Bodies may be invovled in drug-resistance mechanisms. This study showed some interesting findings on the effect of MEKi treatment on RAS expression in cancer cells. 

Some Major points: 

The data on the correlation between the dissolution of P-Bodies and the increased RAS protein expression are not convincing. 

1. Figure 4A. The quality of the imaging need to be improved. The effects of silencing of DDX6, LSM14 on KRAS expression and P-Bodies need to be shown by using immunohistochemical and immunofluorescence analysis. 

2. Author stated that "The silencing of DDX6, LSM14, and 4E-T, which resulted in dissolution of P- Bodies, was correlated with a slight increased RAS expression (Fig. 4F-H and Supplemen tary Fig. S5) but also associated with important lethality that might play an indirect role on RAS expression (data not shown)."

Please show the data to support the statement of the "lethality and the role on RAS expression. 

3. It seems that the MEKi treatment induces feedback-loops (via P-bodies) to induce RAS expression. However, more experimental data need to be shown to support that the MEK inhibitors had significant effects on the P-bodies and on the cell proliferation/growth, the downstream effectors of the MEK inhibitors. 

4. Disscussion section, need extensively assessment on the mechanisms by which MEKi treatment induces feedback-loops (via P-bodies) to induce RAS expression.

Author Response

This study is to investigate whether the induction of RAS proteins by the MEK inhibitors (MEKi) is associated with a change in Processing Body (P-body) biocondensates in cancer cells. The authors showed that both PD184352 and trametinib can decrease the size and number of P-Bodies, which indicating that P-Bodies may be invovled in drug-resistance mechanisms. This study showed some interesting findings on the effect of MEKi treatment on RAS expression in cancer cells. 

We thank the reviewer for these comments and nice suggestions, and we modified the manuscript accordingly in order to improve the quality of the work.

Some Major points: 

The data on the correlation between the dissolution of P-Bodies and the increased RAS protein expression are not convincing. 

  1. Figure 4A. The quality of the imaging needs to be improved. The effects of silencing of DDX6, LSM14 on KRAS expression and P-Bodies need to be shown by using immunohistochemical and immunofluorescence analysis. 

We thank the reviewer for these comments. In order to improve the figure, we have now divided it into two parts, one for overexpression and one for inhibition. As suggested by the reviewer, in the new Figure 5, we added an immunofluorescence panel and quantification of the expression of KRAS under the conditions of siDDX6 and LSM14A. We also observed an absence of P bodies in the presence of siRNA against siDDX6 and LSM14A in association with increased expression of KRAS (new Figure 5 and supplemental figures).

  1. Author stated that "The silencing of DDX6, LSM14, and 4E-T, which resulted in dissolution of P- Bodies, was correlated with a slight increased RAS expression (Fig. 4F-H and Supplemen tary Fig. S5) but also associated with important lethality that might play an indirect role on RAS expression (data not shown)." Please show the data to support the statement of the "lethality and the role on RAS expression. 

It is a pivotal point and we agree with the reviewer. As previously published, DDX6 and Pbody components are essential genes for cell proliferation, as described in recent publications (ex: Hardy SD, et al. Regulation of epithelial-mesenchymal transition and metastasis by TGF-β, P-bodies, and autophagy. Oncotarget. 2017 Oct 17;8(61):103302-103314. doi: 10.18632/oncotarget.21871) and also shown in DepMap: The Cancer Dependency Map Project, where DDX6 is noted as an essential gene. Taken altogether, our results are consistent with these findings. Nevertheless, as suggested by the reviewer, we added in supplementary a panel showing the loss of confluence in siDDX6 and siLSM14 conditions, which is consistent with the cell shrinkage observed by confocal studies (Figure 5).

  1. It seems that the MEKi treatment induces feedback-loops (via P-bodies) to induce RAS expression. However, more experimental data need to be shown to support that the MEK inhibitors had significant effects on the P-bodies and on the cell proliferation/growth, the downstream effectors of the MEK inhibitors. 

We apologize for the possible misunderstanding, but, in this article, we have shown in Figure 1,2,3,8 that MEKi block ERK phosphorylation. As for the downstream effects, in supplemental Figure 1 we show cell cycle arrest after 24 hours and in Figure 7 we show a decrease in cell number and that these results are associated with p-body disassembly.

  1. Disscussion section, need extensively assessment on the mechanisms by which MEKi treatment induces feedback-loops (via P-bodies) to induce RAS expression.

In agreement with the reviewer, we have added two parts in the discussion.

The first is based on the bioinformatic analysis of the possible effect of MEKi on direct or indirect P-body targets based on the article by Ünal EB [Ünal EB, Uhlitz F, Blüthgen N. A compendium of ERK targets. FEBS Lett. 2017 Sep;591(17):2607-2615. doi: 10.1002/1873-3468.12740. Epub 2017 Jul 25. PMID: 28675784.]. In the second part, we now mention two studies showing selective enhancement of translation of specific mRNA by effects on the EIF4 complex by MEKi, as well as the possible dual effects of MEKi on p-body and translation.

Reviewer 3 Report

This manuscript by Vidal-Cruchez et al, presents the concept that resistance to MEK inhibitors is mediated, in part by elevated RAS protein levels caused by the release of RAS mRNA sequestered in p-bodies due to p-body dissolution.  This is an interesting hypothesis and may be the first to attempt to define a role for p-bodies in the regulation of RAS translation.  Targeted inhibitor escape due to unexpected compensatory molecular mechanisms is an important area of study with considerable implications for patient care.

The authors show an increase in K and N-RAS protein levels in some cases after MEK inhibition and effects on p-bodies when cells are treated with MEKi.  However,  

Overall, many of the conclusions are not strongly supported by the data presented, which may be vulnerable to over interpretation.  This reduces enthusiasm for the study.

1. Fig1A:   In the raw data figure, there does not appear to be much change in MEL501 cells and only with PD18 for K-RAS and N-RAS in A549.  So the representative figure does not correspond well to the quantification provided (1B and 1C) or the description in the text. 

2.  The RNA level figure in the supplementary data probably should be shown in the main text.

3.  In Figure 2A the levels of K-Ras protein appear much lower in the Actinomycin D treated samples.  Even at time 0. The quantified graph does not seem to match well to the raw data.  There is no control for ActD treatment without PD18.

4.  The Polysome shift is not hugely convincing.  It looks more like a general change in composition rather than a specific effect on RAS mRNA.   

5.  It I stated that “This MEKi-induced decrease was already significant after 8h, indicating that P-body dissolution kinetics correlate with the time course of KRAS and NRAS overexpression  (Line 237)

But there does not seem to be any change in the levels of RAS protein between hours 0 and 8 in Figure 2.

6.  In Figure 3 it is MEL501 cells that seem to show the biggest change in p-bodies.  Yet in Figure 1 it is these cells that do not appear to show any change in RAS protein levels.  Is there an explanation for this apparent discrepancy?

7.  In Figure 4F the Si control seems to have a noticeable effect on N-RAS expression, which makes interpretation of the figure difficult.

8. In Figure 6, the P body changes seem very subtle and not especially convincing

9.  In Figure 7 the RAS siRNAs do not seem very specific, as the K-RAS siRNA appears to act on N-RAS and vice versa.   Again, this complicates interpretation.

Author Response

This manuscript by Vidal-Cruchez et al, presents the concept that resistance to MEK inhibitors is mediated, in part by elevated RAS protein levels caused by the release of RAS mRNA sequestered in p-bodies due to p-body dissolution.  This is an interesting hypothesis and may be the first to attempt to define a role for p-bodies in the regulation of RAS translation.  Targeted inhibitor escape due to unexpected compensatory molecular mechanisms is an important area of study with considerable implications for patient care.

The authors show an increase in K and N-RAS protein levels in some cases after MEK inhibition and effects on p-bodies when cells are treated with MEKi.  However,  overall, many of the conclusions are not strongly supported by the data presented, which may be vulnerable to over interpretation.  This reduces enthusiasm for the study.

We thank the reviewer for these comments and the different suggestions made, which has helped to improve the quality of the article.

  1. Fig1A:   In the raw data figure, there does not appear to be much change in MEL501 cells and only with PD18 for K-RAS and N-RAS in A549.  So the representative figure does not correspond well to the quantification provided (1B and 1C) or the description in the text. 

We replaced the blot for A549 as suggested. For Mel501, the expression of RAS is rather low and we obtained the same results.

  1. The RNA level figure in the supplementary data probably should be shown in the main text.

 Thank you for this important suggestion. Therefore, the RNA levels are now in the main figure 2 as suggested.

  1. In Figure 2A the levels of K-Ras protein appear much lower in the Actinomycin D treated samples.  Even at time 0. The quantified graph does not seem to match well to the raw data.  There is no control for ActD treatment without PD18.

We thank the reviewer for these comments, and we have identified the possible misunderstanding related to this panel. Since MEKi does not appear to affect mRNA levels of KRAS, the goal of these experiments was to show that the increased protein expression of KRAS occurs even in the absence of neotranscription, arguing for a posttranscriptional mechanism. However, since this observation has been published elsewhere and this panel is not necessary for the understanding of the article, we removed it.

  1. The Polysome shift is not hugely convincing.  It looks more like a general change in composition rather than a specific effect on RAS mRNA.   

We totally agree with the reviewer that the line drawn on the profile was not the best option to show the shift to the polysomal fraction. For this purpose, we added the % of mRNA in the heavy polysomal fraction to better illustrate our results. The DO profile showed a similar profile in the untreated or treated condition, whereas the about 15% KRAS and NRAS mRNA were shifted to the polysomal fraction. Considering that the mRNA located in this heavy fraction is translated at least two to 10 times, the difference at the end is consistent with the fact that mRNA levels do not change while protein levels are increased. In addition, we now mention in the discussion two studies showing a selective increase in translation of specific mRNA through effects on the EIF4 complex by MEKi, as well as the possible dual effects of MEKi on p-body and translation.

  1. It I stated that “This MEKi-induced decrease was already significant after 8h, indicating that P-body dissolution kinetics correlate with the time course of KRAS and NRAS overexpression”  (Line 237)

But there does not seem to be any change in the levels of RAS protein between hours 0 and 8 in Figure 2.

As previously mentioned, we removed the panel containing the actinomycin D experiments. In addition, we decided to minor our conclusion for this figure as suggested by the general comment of the reviewer: “This MEKi-induced a significant P-body dissolution after 8h. The KRAS and NRAS level changes were progressive with a full increase after 24h.”

  1. In Figure 3 it is MEL501 cells that seem to show the biggest change in p-bodies.  Yet in Figure 1 it is these cells that do not appear to show any change in RAS protein levels.  Is there an explanation for this apparent discrepancy?

We fully agree with this comment. The explanation is that the MEL501 harbor fewer P-bodies in terms of size and number. Thus, the P-bodies are more subject to change, while their capacity to store mRNA appears to be limited.

  1. In Figure 4F the Si control seems to have a noticeable effect on N-RAS expression, which makes interpretation of the figure difficult.

We repeated this experiment with different controls and irrelevant siRNA. It seems that the control without transfection is actually too confluent compared to the transfected cells, leading to a possible bias in interpretation.

  1. In Figure 6, the P body changes seem very subtle and not especially convincing

We agree with the reviewer that the panel was not well presented. So, we decided to show the results in log2 to improve the representation. In complement, we added in the supplementary the representative microscopy images associated with the quantifications.

  1. In Figure 7 the RAS siRNAs do not seem very specific, as the K-RAS siRNA appears to act on N-RAS and vice versa.  Again, this complicates interpretation.

The siRNA of KRAS and NRAS are specific in term of extinction as shown in the figure or in previously articles. [Hobbs GA, et al. Atypical KRASG12R Mutant Is Impaired in PI3K Signaling and Macropinocytosis in Pancreatic Cancer. Cancer Discov. 2020 Jan;10(1):104-123. doi: 10.1158/2159-8290.CD-19-1006] or [Dolgikh N, et al. NRAS-Mutated Rhabdomyosarcoma Cells Are Vulnerable to Mitochondrial Apoptosis Induced by Coinhibition of MEK and PI3Kα. Cancer Res. 2018 Apr 15;78(8):2000-2013. doi: 10.1158/0008-5472.CAN-17-1737].

What we cannot rule out is that downregulation of KRAS causes upregulation of NRAS (either by transcription or translation) and vice versa. Here, we have shown that they can complement each other directly after MEKi treatment and that double invalidation is required to completely block BRAF phosphorylation.
